# Mesoporous Nitrogen-Doped Carbon Support from ZIF-8 for Pt Catalysts in Oxygen Reduction Reaction

**DOI:** 10.3390/nano15020128

**Published:** 2025-01-16

**Authors:** Sangyeup Park, Jong Gyeong Kim, Youngin Cho, Chanho Pak

**Affiliations:** Graduate School of Energy Convergence, Institute of Integrated Technology, Gwangju Institute of Science and Technology, Gwangju 61005, Republic of Korea; rupert915@gm.gist.ac.kr (S.P.); xaso123@gm.gist.ac.kr (J.G.K.); choxx576@gm.gist.ac.kr (Y.C.)

**Keywords:** zeolite imidazolate framework, nitrogen-doped carbon, oxygen reduction reaction, supported catalyst, mesoporous structure

## Abstract

Zeolitic imidazolate framework-8 (ZIF-8) has been extensively studied as a precursor for nitrogen-doped carbon (NC) materials due to its high surface area, tunable porosity, and adjustable nitrogen content. However, the intrinsic microporous structure of the ZIF-8 limits mass transport and accessibility of reactants to active sites, reducing its effectiveness in electrochemical applications. In this study, a soft templating approach using a triblock copolymer was used to prepare mesoporous ZIF-8-derived NC (Meso-ZIF-NC) samples. The hierarchical porous structure was investigated by varying the ratios of Pluronic F-127, NaClO_4_, and toluene. The resulting Meso-ZIF-NC exhibited widespread pore size distribution with an enhanced mesopore (2–50 nm) volume according to the composition of the reaction mixtures. Pt nanoparticles were uniformly dispersed on the Meso-ZIF-NC to form Pt/Meso-ZIF-NC catalysts, which presented a high electrochemical surface area and improved oxygen reduction reaction activity. The study highlights the important role of mesopore structure and nitrogen doping in enhancing catalytic performance, providing a pathway for advanced fuel cell catalyst design.

## 1. Introduction

Environmental concerns have driven the search for cleaner and more efficient energy solutions. Fuel cells have emerged as a promising technology due to their ability to convert directly chemical energy into electrical energy with high efficiency and low environmental impact, producing only water as a byproduct when using hydrogen as a fuel [1,2,3,4]. However, successful application of fuel cell systems requires developing advanced materials that can improve their efficiency and durability while reducing costs [5,6,7]. Recently, nitrogen-doped carbon (NC) support has increased significant attention for its potential to address the above challenges [8,9,10].

NC offers significant advantages, including enhanced catalytic activity due to the creation of active sites by nitrogen atoms, improved electrical conductivity resulting from altered electronic structures, and superior chemical stability by the anchoring effect of nitrogen that ensures durability under harsh operational conditions [11,12,13,14]. To achieve these properties, NC can be prepared through methods such as chemical vapor deposition, pyrolysis of nitrogen-containing precursors, and post-treatment with nitrogen-rich gases like ammonia [15,16,17,18,19,20]. Among these, the adoption of metal-organic frameworks (MOFs) as precursors is particularly effective because they make it easier to produce NC [21,22,23,24,25]. Among the MOF materials, ZIF-8 is especially popular for its stability and high nitrogen content, which makes it ideal for producing nitrogen-rich carbon and has enabled its application in a variety of fields, including gas storage, separation technologies, and catalysis [26,27,28,29,30]. Among these applications, ZIF-8-derived NC has been extensively employed as a non-platinum group metal (non-PGM) catalyst for oxygen reduction reaction (ORR) in fuel cells [31,32,33,34,35,36,37]. The nitrogen species within its structure play a key role by improving the overall electron transfer processes, which directly enhances the efficiency of the ORR. Although ZIF-8-derived carbon has been widely used as a non-platinum catalyst, its application as a carbon support material for supported metal catalysts remains not yet widely utilized.

Using ZIF-8-derived NC as a carbon support offers several benefits [38,39,40,41]. The nitrogen-doped structure improves interactions with catalytic nanoparticles, enhancing their dispersion and stability. This interaction promotes efficient electron transfer, boosting catalytic performance. Additionally, the porous structure of an NC facilitates better mass transport, ensuring reactants can reach active sites successfully [42,43,44]. These features make ZIF-8-derived NC a versatile and promising support material for metal catalysts in fuel cells. However, the microporosity of ZIF-8-derived carbon limits its performance in applications requiring efficient mass transport and active site accessibility. To address this limitation, researchers have focused on creating mesoporous ZIF-8-derived NC. Mesopores improve diffusion efficiency, making it easier for reactants to access active sites and enhancing catalytic reactions [45]. They also enhance the dispersion and integration of catalytic nanoparticles, further boosting performance.

Researchers have explored various methods to develop mesoporous ZIF-8, aiming to overcome the limitations of its microporous structure. The etching method uses acidic or basic solutions like tannic acid to selectively dissolve parts of the ZIF-8 framework, creating mesopores while preserving structural integrity [46]. This approach enhances mass transport and improves accessibility to active sites, making it suitable for electrochemical applications. Another approach involves using 2-aminobenzimidazole as a mixed-linker during synthesis, which introduces hierarchical porosity by partially re-placing the original framework [47]. This method accelerates molecular diffusion and boosts catalytic efficiency. Finally, soft template-assisted synthesis leverages micelle-forming agents to control pore size and structure, offering scalability and uniformity [48,49,50,51]. This technique produces hierarchical materials that significantly improve diffusion and transport properties, making it a versatile solution for various catalytic and energy applications.

This study focuses on overcoming the inherent limitations of microporous ZIF-8-derived carbons by synthesizing mesoporous ZIF-8-derived nitrogen-doped carbon (Meso-ZIF-NC) through a soft-templating approach. Introducing hierarchical mesoporosity significantly enhances mass transport, active site accessibility, and catalytic activity. The resulting Meso-ZIF-NC was used as a support for Pt nanoparticles, achieving uniform dispersion and enhanced interaction with the carbon support. The catalysts demonstrated improved ORR activity, highlighting the role of nitrogen doping and generated mesoporosity. These findings provide valuable insights into the rational design of carbon materials for advanced fuel cell applications.

## 2. Materials and Methods

### 2.1. Synthesis of Mesoporous ZIF-8 Based Nitrogen-Doped Carbon (Meso-ZIF-NC)

The synthesis of mesoporous ZIF-8 begins with the preparation of two separate solutions. In the first step, Pluronic F-127 (Sigma Aldrich, St. Louis, MO, USA) is dissolved in methanol (MeOH, Duksan, Ansan, Republic of Korea) to form Solution 1 (Sol.1), and another identical amount of F-127 is dissolved in 12.5 mL of methanol to form Solution 2 (Sol.2). Next, NaClO_4_ (Samchun, Seoul, Republic of Korea) and toluene (Daejung, Siheung, Republic of Korea) are added to both Sol.1 and Sol.2, followed by stirring for 1 h. To prepare ZIF-8, 2-methylimidazole (0.657 g, Sigma Aldrich, St. Louis, MO, USA) is added to Sol.1, and zinc nitrate hexahydrate (0.565 g, Deajung, Siheung, Republic of Korea) is added to Sol.2. Both solutions are stirred separately for 1 h. After the precursor solutions are ready, Sol.2 is gradually added into Sol.1 under continuous stirring. The resulting mixed solution is stirred for 24 h at 40 °C to allow for the crystallization of mesoporous ZIF-8. The solution was filtered, and the white precipitate was sequentially washed several times with ethanol (EtOH) and MeOH and dried at 60 °C in a vacuum oven for 12 h. The dried white powder was finely ground and subjected to pyrolysis in an Ar atmosphere using a tube furnace. During pyrolysis, the temperature was gradually increased at 5 °C/min until reaching 350 °C, where it was maintained for 2 h. This step effectively removed moisture, volatile components, and functional groups from the precursor, providing a clean foundation for the carbonization process. Subsequently, the temperature was further increased to 900 °C at the same rate of 5 °C per minute and held steady for 2 h. The obtained black powder was denoted as Meso-ZIF-NC.

To explore the influence of the molecular ratios of key components on the mesoporous structure, the preparation was conducted with six different compositions. These compositions are presented in Table 1, and the samples were named Meso-ZIF-NC-*n*, where *n* is 1 to 6, as listed in Table 1.

### 2.2. Preparation of Pt/Meso-ZIF-NC

In a typical synthesis, 20 mg of Meso-ZIF-NC was added to 10 mL acetone. The solution was then bath-sonicated for 1 h and gently stirred for 1 h. Meanwhile, 23.2 mg of hexachloroplatinic acid hexahydrate (H_2_PtCl_6_·6H_2_O) was dissolved in 5 mL of acetone and sonicated for 5 min, then added dropwise to the carbon solution. The mixture was stirred for 2 h and dried at 60 °C in a vacuum oven after evaporating the solvent. The mixture was ground and thermally reduced in 10% H_2_ and balanced N_2_ at 200 °C with a ramp rate of 5 °C/min. After 3 h at 200 °C, it was cooled naturally under Ar.

### 2.3. Characterization of Materials

The morphology and pore architectures were analyzed using field emission scanning electron microscopy (FE-SEM, S4700, Hitachi, Tokyo, Japan) and ultra-high-resolution field emission scanning electron microscopy (UH-FE-SEM, Verios 5 UC; Thermo Fisher Scientific, Waltham, MA, USA). Transmission electron microscopy (TEM; Tecnai G2 F30 S-Twin; FEI Company, Hillsboro, OR, USA) was employed to examine the material’s morphology, and energy-dispersive spectroscopy (EDS) mappings were acquired using the Oxford Ultim Max 65 instrument. Nitrogen (N_2_) adsorption isotherms were used to assess the physical properties of Meso-ZIF-NC, measured at 77 K using a Belsorp MAX instrument (Microtrac MRB, Osaka, Japan). The specific surface area was calculated using the Brunauer–Emmett–Teller (BET) method, while the pore size distribution (PSD) was determined through non-local density functional theory (NLDFT). The structural characteristics of Meso-ZIF-NC were investigated via X-ray diffraction (XRD, SmartLab; Rigaku, Tokyo, Japan), and the surface electron states of its components were analyzed using X-ray photoelectron spectroscopy (XPS, NEXSA; Thermo Fisher Scientific, Waltham, MA, USA). Raman spectroscopy (LabRAM HR Evolution; Horiba, Kyoto, Japan) was performed to examine the I_D_/I_G_ ratio of the Meso-ZIF-NC.

### 2.4. Electrochemical Measurements

Electrochemical evaluation of the catalysts was carried out using a potentiostat (VSP, BioLogic, Seyssinet-Pariset, France) and a three-electrode electrochemical setup. The working electrode was a glassy carbon rotating disk electrode (GC-RDE, ϕ5 mm; Pine Research, Durham, NC, USA), which was polished with 0.05 µm alumina powder and thoroughly rinsed with deionized water. A reversible hydrogen electrode (RHE; Gaskatel, Kassel, Germany) served as the reference electrode, while a platinum coil encased in glass (AFCTR5; Pine Research, Durham, NC, USA) was used as the counter electrode. Catalyst inks were prepared by mixing 5 mg of catalyst with 10 µL of 10 wt% Nafion^®^ solution (EW 1100; Sigma-Aldrich, MO, USA) in 2.49 mL of anhydrous EtOH. The catalyst loading on the working electrode was set to 30 µg_Pt_/cm^2^. Electrochemical testing was performed under consistent conditions for all samples. Prior to the ORR activity test, the electrolyte was purged with N_2_ and activated (30 cycles at 100 mV/s, 0.05–1.0 V) in 0.1 M HClO_4_. Cyclic voltammetry (CV) measurements were conducted from 0.05 to 1.2 V (50 mV/s) in an N_2_-purged electrolyte. Linear sweep voltammetry (LSV) measurements were conducted from 1.2 to 0.1 V (10 mV/s, 1600 rpm) in an O_2_-purged electrolyte. For the CO-stripping voltammetry measurements, the working electrode was immersed in the CO-saturated for 10 min while maintaining the electrode potential at 0.1 V. Excess CO was removed by purging with N_2_ gas for 45 min. Finally, CV measurements were conducted from 0.1 to 1.2 V (20 mV/s) in an N_2_-purged electrolyte.

## 3. Results and Discussion

### 3.1. Key Components in the Synthesis of Meso-ZIF-NC

The synthesis of Meso-ZIF-NC materials involves the use of F-127, NaClO_4_, and toluene as critical components to design the mesoporous structure. The F-127 serves as a structure-directing agent, forming micelles that act as templates for mesopore formation. The NaClO_4_ acts as a micelle-swelling agent, enhancing the interaction between the micelles and the ZIF-8, while toluene further facilitates micelle expansion and structural organization [52,53,54,55]. The materials were synthesized using a soft template approach. As shown in Figure 1, the 2-methylimidazole and zinc nitrate hexahydrate were dissolved in a methanol-based solution containing varying ratios of F-127, NaClO_4_, and toluene. The mixture was processed with controlled crystallization to form mesoporous ZIF-8. This was followed by carbonization to convert the Meso-ZIF-NC as shown in Figure 1.

The pore volume of the Meso-ZIF-NC materials varies significantly depending on the ratios of F-127, NaClO_4_, and toluene. Below, the effects of these variations are discussed in detail in Table 2 and Appendix A. Meso-ZIF-NC-2 was synthesized using F-127, NaClO_4_, and toluene, resulting in a mesoporous structure. In the case of Meso-ZIF-NC-3, the amount of F-127 was doubled compared to Meso-ZIF-NC-2 while maintaining the same ratios of NaClO_4_ and toluene. Despite this increase in F-127 concentration, the mesopore volume did not show a significant enhancement. The lack of effect on mesopore volume can be attributed to the saturation of micelle formation. F-127 acts as a structure-directing agent by forming micelles that serve as templates for mesopore development. However, beyond a certain concentration, additional F-127 does not further increase micelle formation or interaction with the ZIF-8, as the available precursors and solvent interactions become limiting factors. As a result, doubling the F-127 concentration does not significantly impact the mesoporous structure, indicating that the micelle templating effect has reached its maximum efficiency under these conditions.

In the case of Meso-ZIF-NC-4, the amount of NaClO_4_ was doubled compared to Me-so-ZIF-NC-2, while maintaining the same ratios of F-127 and toluene. This increase in NaClO_4_ resulted in a noticeable enhancement in mesopore volume, highlighting its critical role in mesoporous structure formation (Table 2, Appendix A). The observed increase in mesopore volume can be explained by the role of NaClO_4_ as a micelle-swelling agent. Higher concentrations of NaClO_4_ strengthen the interaction between the ZIF-8 and the micelle templates formed by F-127. This stronger interaction promotes the expansion of micelle cores, leading to the development of more accessible mesopores. Furthermore, the enhanced swelling effect creates a more open and interconnected mesoporous structure, which contributes to the overall increase in pore volume. Thus, the addition of NaClO_4_ beyond the baseline amount effectively amplifies the hierarchical porosity of the material.

Meso-ZIF-NC-5 was synthesized with double the amount of toluene compared to Meso-ZIF-NC-2 while keeping the ratios of F-127 and NaClO_4_ constant. The results show that increasing the toluene concentration leads to a decrease in mesopore volume, indicating that excessive toluene negatively impacts the mesoporous structure. The reduction in pore volume can be attributed to the destabilizing effect of excessive toluene on micelle formation. Toluene, as a co-solvent, plays a role in expanding the micelle cores during the synthesis process. However, when the concentration of toluene is too high, it causes overexpansion and destabilization of the micelles, leading to structural collapse during the crystallization of ZIF-8. This results in the formation of fewer or poorly defined mesopores. Additionally, excessive toluene may disrupt the balance of interactions between F-127, NaClO_4_, and ZIF-8, further hindering the development of an optimal mesoporous framework.

Meso-ZIF-NC-6 was synthesized by doubling the molar amounts of F-127, NaClO_4_, and toluene compared to Meso-ZIF-NC-2. This adjustment resulted in a significant increase in the mesopore volume, particularly in the range of 2–50 nm, demonstrating the importance of the combined effects of these three components in enhancing mesoporosity. The increase in pore volume can be explained by the synergistic roles of F-127, NaClO_4_, and toluene during the synthesis process. Doubling the amount of F-127 increases the number of micelle templates, while the higher concentration of NaClO_4_ enhances the swelling of the micelle cores. Meanwhile, the increased toluene concentration facilitates further micelle expansion and stabilizes the formation of a hierarchically porous structure. These effects intensify the interaction between the ZIF-8 and the templating agents, leading to a more robust and interconnected mesoporous network. This result highlights that the simultaneous optimization of F-127, NaClO_4_, and toluene concentrations is vital for maximizing the mesopore volume and achieving a highly mesoporous ZIF-derived nitrogen-doped carbon material.

### 3.2. Materials Characteristics

The STEM images (Figure 2a–c) showed that Meso-ZIF-NC-1 has the least mesoporous structure, appearing compact and dense due to the absence of F-127, NaClO_4_, and toluene during its synthesis. In contrast, Meso-ZIF-NC-5 exhibits an intermediate mesoporous structure, as shown by its slightly more open framework. This improvement is due to the inclusion of these structure-directing agents. Meso-ZIF-NC-6 demonstrates the most developed mesoporous structure, with clear hierarchical porosity, as it was synthesized using optimized amounts of F-127, NaClO_4_, and toluene. The SEM images (Figure 2d–f) further confirm these trends. Meso-ZIF-NC-1 shows a smooth, compact surface, while Meso-ZIF-NC-5 has a moderately porous structure. Meso-ZIF-NC-6 exhibits a rough and highly porous surface, consistent with its superior mesoporosity. The STEM and SEM images of other Meso-ZIF-NC samples further show differences in porosity and surface characteristics, reflecting the impact of synthesis conditions (Appendix A). The pore size distribution curves (Figure 2h) reveal distinct differences in mesopore contributions among Me-so-ZIF-NC-1, Meso-ZIF-NC-5, and Meso-ZIF-NC-6. Meso-ZIF-NC-1 exhibits a minimal contribution from mesopores, with most of its porosity arising from micropores. This is consistent with its synthesis conditions, where no structure-directing agents were used, resulting in a compact and dense structure. In comparison, Meso-ZIF-NC-5 displays a noticeable increase in mesopore contributions, particularly in the 2–50 nm range, as shown by the expansion of the pore size distribution. This suggests that the partial use of structure-directing agents during synthesis enabled the formation of mesopores, contributing to a more open framework than Meso-ZIF-NC-1. Meso-ZIF-NC-6, synthesized under optimized conditions, demonstrates the most pronounced peak in the 2–50 nm range. This trend is further supported by the nitrogen adsorption/desorption isotherms (Figure 2g), where Meso-ZIF-NC-6 exhibits the highest nitrogen uptake, correlating with its larger mesoporous volume. Notably, significant contributions are observed from both smaller mesopores (2–10 nm) and larger mesopores (10–50 nm), reflecting a well-developed hierarchical pore structure as shown in red box of Figure 2h.

The XRD patterns (Figure 3a) for Meso-ZIF-NC-1, Meso-ZIF-NC-5, and Meso-ZIF-NC-6 exhibit similar diffraction peaks, indicating that the crystalline structure of the materials remains consistent across the samples. All samples exhibit a broad peak around 26°, corresponding to the (002) plane of graphitic carbon. This suggests that the materials share a disordered stacking of graphitic layers. The consistent position of the (002) peak (~26°) across all samples indicates a similar interlayer spacing, reflecting comparable graphitic domains. Similarly, a weak and broad peak near 45–46°, corresponding to the (100) plane of graphitic carbon, is present in all three samples. This peak represents the in-plane arrangement of sp^2^ carbon atoms and highlights the partially disordered nature of the carbon frameworks derived from ZIF-8. The relative intensity and broadness of the (100) peaks were consistent among the samples, further supporting the structural similarity of the carbon matrices. The same trend was observed in other Meso-ZIF-NC samples, highlighting the uniformity in structural characteristics across the series (Appendix A). The Raman spectra of the Meso-ZIF-NC (Figure 3b) reveal their carbon structures. The I_D_/I_G_ ratios, representing the intensity of the D band (1350 cm⁻^1^) relative to the G band (1580 cm⁻^1^), were calculated as 1.28, 1.43, and 1.18, respectively. Since all I_D_/I_G_ values exceed 1, it confirms that all three samples are predominantly composed of amorphous carbon, characterized by significant structural defects and disorder. Amorphous carbon is characterized by significant structural defects and a lack of long-range order, which can facilitate catalytic activity by exposing more reactive sites for chemical interactions. The defective carbon framework offers enhanced accessibility to active sites, making these materials particularly advantageous for catalytic and electrochemical applications. Other Meso-ZIF-NC carbons also exhibit the same trend, further supporting the notion that the structural defects and disorder are consistent across the samples (Appendix A).

Figure 4 provides an in-depth analysis of the structural and compositional characteristics of Pt/Meso-ZIF-NC catalysts synthesized with varying mesoporous carbon supports. The STEM image (Figure 4a) shows the uniform distribution of Pt nanoparticles embedded within the porous carbon framework, highlighting the effectiveness of the synthesis method. Elemental mapping (Figure 4b and Appendix A) further confirms the homogeneous dispersion of carbon, nitrogen, oxygen, and platinum across the material. The nitrogen doping in the carbon matrix is particularly noteworthy, as it enhances the interaction between the Pt nanoparticles and the carbon support, thereby improving the stability and catalytic performance of the material.

The XRD patterns (Figure 4c) confirm the crystalline structure of Pt nanoparticles supported on Meso-ZIF-NC-1, Meso-ZIF-NC-5, and Meso-ZIF-NC-6. Distinct diffraction peaks corresponding to Pt (111), (200), and (220) crystal planes are observed in all three samples, indicating successful deposition of platinum on the carbon frameworks. Using the Scherrer equation, the crystallite sizes of Pt were calculated to be 4.6 nm, 4.4 nm, and 3.8 nm for Pt/Meso-ZIF-NC-1, Pt/Meso-ZIF-NC-5, and Pt/Meso-ZIF-NC-6, respectively. The smallest crystallite size of Pt/Meso-ZIF-NC-6 can be attributed to the highly developed mesoporous structure formed under optimized synthesis conditions. The mesoporous framework restricts the growth of Pt nanoparticles, resulting in smaller crystallites. This trend is further supported by STEM images (Appendix A), which clearly show the uniform dispersion of Pt nanoparticles within the mesoporous structure.

The XPS survey spectra (Figure 4d) and the elemental composition table reveal the consistent incorporation of key elements across Pt/Meso-ZIF-NC-1, Pt/Meso-ZIF-NC-6, and Pt/Meso-ZIF-NC-6. All samples exhibit the expected peaks for C 1s, N 1s, O 1s, Zn 2p, and Pt 4f, indicating the successful synthesis of NC materials and the deposition of Pt. Additionally, the XPS data focusing solely on NC materials is provided in the Appendix A, further validating the elemental composition and nitrogen doping. The atomic percentages of carbon, nitrogen, and oxygen across the three samples show only minor variations (Appendix A). Carbon levels range from 75.49% to 81.3%, with Meso-ZIF-NC-6 exhibiting the highest value, likely due to the optimized synthesis conditions leading to more carbonized frameworks. Nitrogen content is relatively stable across the samples, ranging from 7.32% to 10.91%, confirming the uniform nitrogen doping into the carbon matrix during pyrolysis. Oxygen content decreases slightly from Pt/Meso-ZIF-NC-1 (10.23%) to Pt/Meso-ZIF-NC-6 (6.74%), which may reflect reduced oxygen functionalities due to the higher degree of carbonization in the mesoporous structures. Zn and Pt contents are consistent with the materials’ synthesis conditions, with Zn ranging from 1.26% to 2.5% and Pt from 1.22% to 4.47%. Pt/Meso-ZIF-NC-1 exhibits the highest Pt content, which may be attributed to less effective dispersion of platinum nanoparticles in the absence of mesoporous structure. In contrast, Pt/Meso-ZIF-NC-6 shows a slightly lower Pt content due to the well-dispersed distribution of nanoparticles within its hierarchical pores.

### 3.3. Electrochemical Characterization

In the CV analysis (Figure 5a), Pt/Meso-ZIF-NC-6 exhibited the highest capacitance among all the samples. This result can be attributed primarily to the large mesopore volume of Pt/Meso-ZIF-NC6, which enhances the accessibility of the electrolyte to the electrode surface. The highly developed mesoporous structure increases the effective surface area, allowing for greater charge storage capacity. This combination of structural and compositional features explains the superior capacitive behavior of Pt/Meso-ZIF-NC-6. The CO-stripping results (Figure 5b) reveal trends similar to those observed in the CV measurements. Pt/Meso-ZIF-NC-6 demonstrated a pronounced CO desorption peak, followed by a capacitive response that is significantly higher than those of the other samples. The higher capacitive behavior is consistent with the large mesopore volume and well-dispersed Pt nanoparticles of Pt/Meso-ZIF-NC-6. These structural characteristics allow for a more uniform CO adsorption and desorption process, maximizing the electrochemical activity. Additionally, a notable feature of the CO-stripping results for Meso-ZIF-NC-based catalysts is the splitting of the CO oxidation peak. This splitting may be attributed to the residual Zn present in the carbon matrix. Zn, introduced during the synthesis of Meso-ZIF-NC, can alter the local electronic environment and interaction between CO and Pt nanoparticles, leading to the observed peak splitting. This phenomenon highlights the influence of Zn on the electrochemical behavior of the catalysts and suggests that its presence could modulate the CO adsorption and desorption characteristics. In the LSV analysis for the ORR (Figure 5c), Pt/Meso-ZIF-NC-5 demonstrated the highest half-wave potential, indicating the best catalytic performance among the samples. In contrast, Pt/Meso-ZIF-NC-1 exhibited the lowest ORR activity. This can be attributed to the distribution of Pt nanoparticles on the surface due to insufficient mesopores or structural constraints, which leads to a reduced electrochemical surface area and poor catalytic performance. On the other hand, while Pt/Meso-ZIF-NC-6 has the most well-developed mesoporous structure, the Pt nanoparticles are more likely to be embedded deep within the pores. This reduces their accessibility for oxygen reduction, slightly limiting the ORR activity compared to Pt/Meso-ZIF-NC-5. The electrochemical analysis highlights the significant role of mesopore volume, Pt dispersion, and carbon support properties in determining the performance of Pt/Meso-ZIF-NC catalysts. While Pt/Meso-ZIF-NC-6 benefits from its large mesopore volume and enhanced capacitance, Pt/Meso-ZIF-NC-5 achieves the best catalytic performance due to its optimal structural and compositional characteristics. These findings underline the importance of fine-tuning the synthesis conditions to balance mesopore accessibility and Pt nanoparticle dispersion.

## 4. Conclusions

This study successfully synthesized Meso-ZIF-NC using a triblock copolymer-assisted soft-templating approach. The synthesis process involved using F-127, NaClO_4_, and toluene as critical components to tailor the mesoporous structure. F-127 served as a structure-directing agent, forming micelles that acted as templates for mesopore formation. NaClO_4_ functioned as a micelle-swelling agent, enhancing the interaction between the micelles and the ZIF-8 framework, while toluene facilitated further micelle expansion and structural organization. The materials were synthesized by dissolving 2-methylimidazole and zinc nitrate hexahydrate in a methanol-based solution containing varying ratios of F-127, NaClO_4_, and toluene. The mixture was stirred to ensure homogeneous interaction and then subjected to controlled crystallization to form mesoporous ZIF-8, followed by carbonization to convert the material into Meso-ZIF-NC.

The mesoporous structure of Meso-ZIF-NC effectively addressed the limitations of microporosity in conventional ZIF-8-derived carbons, improving mass transport and access to active sites. Additionally, Pt nanoparticles were evenly distributed on the Meso-ZIF-NC, resulting in catalysts with strong electrochemical performance. The Pt/Meso-ZIF-NC catalysts showed improved activity in the ORR and efficient use of Pt, highlighting the benefits of the mesoporous architecture and nitrogen doping. These results demonstrate the potential of Meso-ZIF-NC as a promising support material for advanced fuel cell catalysts. Future research will focus on evaluating the long-term stability of the catalysts and testing their performance in membrane electrode assembly configurations, further advancing their applicability in practical fuel cell systems.

## Figures and Tables

**Figure 1 nanomaterials-15-00128-f001:**
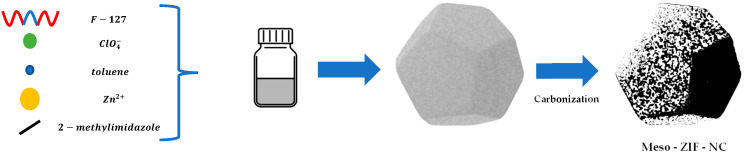
Schematic procedures for synthesis of Meso-ZIF-NC.

**Figure 2 nanomaterials-15-00128-f002:**
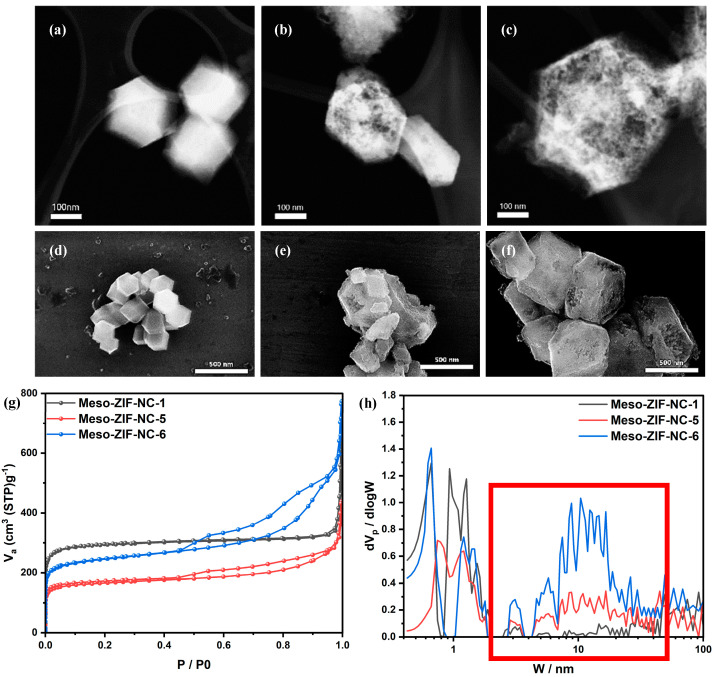
STEM, SEM images of (**a**,**d**) Meso-ZIF-NC-1, (**b**,**e**) Meso-ZIF-NC-5, (**c**,**f**) Meso-ZIF-NC-6, (**g**) N_2_ adsorption/desorption isotherms of Meso-ZIF-NC, and (**h**) pore size distribution curves of Meso-ZIF-NC analyzed by NLDFT method.

**Figure 3 nanomaterials-15-00128-f003:**
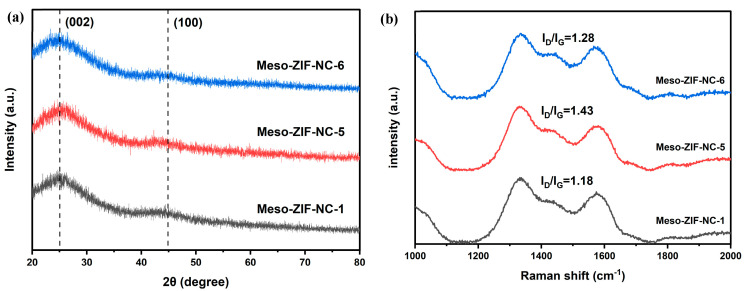
(**a**) XRD patterns and (**b**) Raman spectra of Meso–ZIF–NC-1, Meso-ZIF-NC-5 and Meso-ZIF-NC-6.

**Figure 4 nanomaterials-15-00128-f004:**
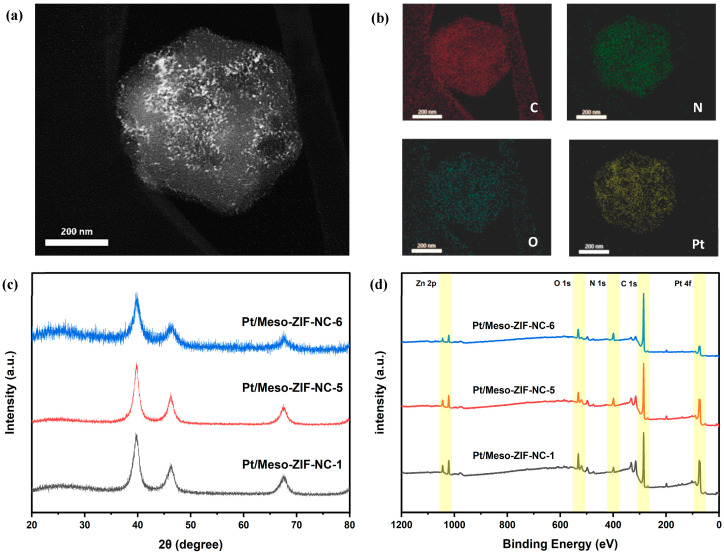
(**a**) STEM image and (**b**) Elemental mapping images of Pt/Meso-ZIF-NC-5. (**c**) XRD patterns and (**d**) XPS full spectrum of Pt/Meso-ZIF-NC-1, Pt/Meso-ZIF-NC-5, and Pt/Meso-ZIF-NC-6.

**Figure 5 nanomaterials-15-00128-f005:**
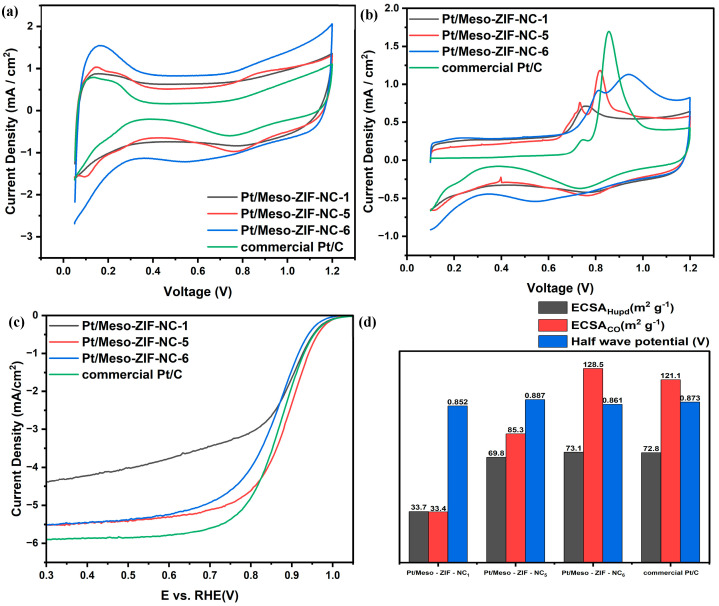
(**a**) CV, (**b**) CO stripping, and (**c**) LSV measurement results in 0.1 M HClO_4_. CV and CO stripping were performed in an N_2_-saturated atmosphere, while LSV was conducted in an O_2_-saturated atmosphere at a rotation rate of 1600 rpm. The electrochemical parameters derived from (**a**–**c**) are summarized in (**d**).

**Table 1 nanomaterials-15-00128-t001:** Reagent compositions for Meso-ZIF-NC synthesis.

Sample	F-127 (mol)	NaClO_4_ (mol)	Toluene (mol)
Meso-ZIF-NC-1	0	0	0
Meso-ZIF-NC-2	3.17 × 10^−5^	8.54 × 10^−3^	3.76 × 10^−3^
Meso-ZIF-NC-3	6.34 × 10^−5^	8.54 × 10^−3^	3.76 × 10^−3^
Meso-ZIF-NC-4	3.17 × 10^−5^	0.017	3.76 × 10^−3^
Meso-ZIF-NC-5	3.17 × 10^−5^	8.54 × 10^−3^	7.52 × 10^−3^
Meso-ZIF-NC-6	6.34 × 10^−5^	0.017	7.52 × 10^−3^

**Table 2 nanomaterials-15-00128-t002:** Specific surface area, total pore volume, micropore, mesopore, and macropore volume of Meso-ZIF-NC. The pore volumes of Meso-ZIF-NC were obtained by integrating PSD data under 100 nm for interparticle porosity.

Sample	S_BET_ ^a^ (m^2^/g)	V_p_ ^b^ (cm^3^/g)	V_micro_ ^c^ (cm^3^/g)	V_meso_ ^d^ (cm^3^/g)	V_macro_ ^e^ (cm^3^/g)
Meso-ZIF-NC-1	1042	0.561	0.447	0.063	0.051
Meso-ZIF-NC-2	941	0.826	0.356	0.393	0.077
Meso-ZIF-NC-3	808	0.787	0.307	0.363	0.117
Meso-ZIF-NC-4	883	0.928	0.319	0.492	0.117
Meso-ZIF-NC-5	589	0.49	0.225	0.211	0.054
Meso-ZIF-NC-6	873	0.916	0.318	0.513	0.085

^a^ Specific surface area; ^b^ Total pore volume; ^c^ Micropore volume (under 2 nm); ^d^ Mesopore volume (2~50 nm); ^e^ Macropore volume (50~100 nm).

## Data Availability

Data are contained within the article and Appendix A.

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
