# Peer review of "Mesoporous Nitrogen-Doped Carbon Support from ZIF-8 for Pt Catalysts in Oxygen Reduction Reaction"

_nanomaterials, 2025, doi:10.3390/nano15020128_

Round 1
Reviewer 1 Report
Comments and Suggestions for Authors
1. In Fig.4b, Please add a ruler to the diagram.
2. What is the cycling performance of the catalyst? Please add
3. Although XPS has been tested in this paper, the data analysis is not sufficient. Please provide the analysis data graph of each element.
4. What is the particle size of platinum? Is it possible to analyze the size distribution of platinum in each sample.
Please refer to and cite these recent papers to revise the manuscript, such as Batteries & Supercaps 2024, 7, e202481102; Next Energy, 2025,6: 100191
Comments on the Quality of English Language
good
Reviewer 2 Report
Comments and Suggestions for Authors
In this work, the authors reported the synthesis of mesoporous ZIF-8-derived nitrogen-doped carbon (Meso-77 ZIF-NC) through a soft-templating approach and their application as an efficient support for Pt catalyst toward the oxygen reduction reaction (ORR). The synthesis conditions were carefully adjusted to obtain the best ORR performance. Overall, this is very interesting work and the manuscript was written in a straight-forward manner. The findings have potential implications for the development of electrocatalysts and their support materials. I believe this work is appropriate for the journal Nanomaterials. However, addressing several minor issues would further improve the clarity and quality of the manuscript. The detailed comments below may be helpful to the authors in this regard.
1. Figure 4b, in addition to the elemental mapping images, it is suggested that the authors also provide the EDS spectra (in the Supplementary Information) to support the presence of each element involved.
2. Recent works about ORR are suggested to be referenced in the Introduction (e.g., DOI: 10.1002/inf2.12608).
3. Figure 4d, the authors only showed the XPS full spectra. How about the high-resolution spectra for certain elements such as Pt, C, and N? They may be able to offer more information about any difference in the electronic structure of the catalysts.
4. Related works on MOF-derived materials can be referred to in the Introduction (doi: 10.1002/sus2.34).
5. Figure 5c, it is recommended that the authors include the rotation rate in the caption, such that the data presented is meaningful for a comparison.
6. How were the he ID/IG ratios obtained? Were they the ratio of the intensity of the ID and IG peaks?
7. There are two Table 1 in the manuscript. These tables should be properly numbered.
